# Study on the Photosynthetic Physiological Responses of Greenhouse Young Chinese Cabbage (*Brassica rapa* L. *Chinensis Group*) Affected by Particulate Matter Based on Hyperspectral Analysis

**DOI:** 10.3390/plants14101479

**Published:** 2025-05-15

**Authors:** Lijuan Kong, Siyao Gao, Jianlei Qiao, Lina Zhou, Shuang Liu, Yue Yu, Haiye Yu

**Affiliations:** 1College of Engineering and Technology, Jilin Agricultural University, Changchun 130118, China; konglijuan@jlau.edu.cn (L.K.); 20240295@mails.jlau.edu.cn (S.G.); zhoulina@jlau.edu.cn (L.Z.); 2College of Horticulture, Jilin Agricultural University, Changchun 130118, China; qiaojianlei@jlau.edu.cn (J.Q.); lshuang@jlau.edu.cn (S.L.); yyue@jlau.edu.cn (Y.Y.); 3College of Biological and Agricultural Engineering, Jilin University, Changchun 130012, China

**Keywords:** particulate matter, greenhouse, young Chinese cabbage, hyperspectral, modeling

## Abstract

Particulate matter affects both the light environment and air quality in greenhouses, obstructing normal gas exchange and hindering efficient physiological activities such as photosynthesis. This study focused on young Chinese cabbage (*Brassica rapa* L. *Chinensis Group*) in a greenhouse at harvest time, monitoring and comparing hyperspectral information, net photosynthetic rate, and microscopic leaf structure under two conditions: a quantitative artificial particulate matter environment and a healthy environment. Based on microscopic results combined with spectral responses and changes in photosynthetic physiological information, it is believed that particulate matter enters plant cells through stomata. Through retention and transport pathways, it disrupts the membrane structure, organelles, and other components of plant cells, resulting in adverse effects on the plant’s physiological functions. The study analyzed the mechanisms by which particulate matter influences the photosynthesis, spectral characteristics, and physiological responses of young Chinese cabbage. Physiological Reflectance Index (PRI), Modified Chlorophyll Absorption Ratio Index (MCARI), spectral red-edge position (λr), and spectral sensitive bands were used as spectral feature variables. Through cubic polynomial and 24 combinations of spectral preprocessing and modeling methods, an inversion model of spectral features and net photosynthetic rate was established. The optimal combination of spectral preprocessing and modeling methods was finally selected as SG + SD + PLS + MSC, which consists of Savitzky-Golay smooth (SG), second derivative (SD), partial least squares (PLS), and multiplicative scatter correction (MSC). The coefficient of determination (R^2^) of the model is 0.9513. The results indicate that particulate matter affects plant photosynthesis. The SG + SD + PLS + MSC combination method is relatively advantageous for processing the photosynthetic spectral physiological information of plants under the influence of particulate matter. The results of this study will deepen the understanding of the mechanisms by which particulate matter affects plants and provide a reference for the physiological information inversion of greenhouse vegetables under particulate matter pollution.

## 1. Introduction

The sunlight greenhouse enables year-round production of facility vegetables and promotes the development of agricultural modernization. When studying the growth environment of facility vegetables, people often focus on factors such as light, temperature, soil [1], and nutrients but tend to overlook the negative impact of particulate matter on both the air and light environment inside and outside the greenhouse [2]. Particulate pollution results in an insufficient effective light environment and poor air quality within the greenhouse, which severely affects the yield and quality of greenhouse vegetables [3]. Young Chinese cabbage (*Brassica rapa* L. *Chinensis Group*), which is rich in nutrients, is widely grown in China and serves as an important source of greenhouse vegetables in the autumn and winter seasons. During the harvest period, the leaves of young Chinese cabbage are wavy and wrinkled, with fine hairs on the leaf surface. This leaf characteristic makes it easier for particulate matter to be adsorbed and enter the leaf stomata, affecting gas exchange, respiration, and photosynthesis, which in turn reduces the net photosynthetic rate and biomass accumulation of greenhouse-grown young Chinese cabbage, hindering the production of high-quality, high-yield greenhouse vegetables [4]. Therefore, studying the effects of particulate matter on the photosynthetic physiology of plants in such environments is of significant importance for guiding agricultural production and scientific research.

In certain specific seasons, such as sandstorms in northern China during spring, soil wind erosion and dust suspension, coal combustion emissions in autumn and winter [5], and haze weather [6], the concentration of environmental particulate matter tends to be high, resulting in a weak light environment [7]. The suspended particulate matter both inside and outside the greenhouse, once adsorbed onto plant leaves, can lead to changes in the plant’s physiological and biochemical processes. These changes manifest as leaf yellowing, premature aging, slow growth, and yield reduction. Long-term exposure to particulate matter is detrimental to the accumulation of biomass, improvement of quality, and morphological development of plants [8,9]. Miguel Izquierdo-Díaz and colleagues studied the impact of particulate air pollution on the fresh consumption of lettuce, showing that the concentration of particulate matter in the air was positively correlated with the amount of particulate matter deposited on lettuce leaves, emphasizing the importance of washing the leaves before consumption [10]. Marie Lhotská and colleagues investigated the adverse effects of dust particles on photosynthetic parameters, such as net photosynthetic rate and transpiration rate, in lettuce leaves. Their research indicated that particulate matter can induce oxidative stress responses in plants, especially highlighting the harmful effects of heavy metals present in particulate matter on the photosynthetic physiological functions of plants [11].

Plants can effectively capture particulate matter, reduce atmospheric particulate pollution, and purify the air. However, the ability of plants to capture atmospheric particulate matter varies, which is closely related to the surface characteristics of the plant, such as trichome density [12], stomatal density [13], leaf surface roughness [13], and leaf shape [14,15,16]. Research by Pei-Pei Gao and colleagues showed that cabbage leaves regulate the absorption of Pb from atmospheric PM2.5 through both stomata and trichomes, determining the proportion of Pb absorbed by the leaf surface in the guard cells and mesophyll cells’ subcellular spaces [17,18]. Taesik Go and colleagues conducted experiments using Digital Inline Holographic Microscopy (DIHM) to study the deposition and movement of PM particles on the surface of leaves [19]. Jihwan Kim and colleagues compared the three-dimensional deposition movement of PM near the microstructures of trichomes in different plants and, based on simulation models, verified that the bending microstructure of leaf trichomes enhances the electric field strength near the trichomes, thus increasing their ability to capture particulate matter [20].

The correlation between particulate matter and plant physiological response has attracted attention from many scholars, with spectral technology often being used for monitoring and analyzing the photosynthetic physiological ecology of plants under the influence of particulate matter [10,21,22]. Spectral diagnosis is based on the response characteristics of plants to different spectral bands. The principle is that when plants are in different environmental conditions, physiological information such as water content, chlorophyll content, and net photosynthetic rate within the plants will change [23]. Specific wavelengths of the visible light spectrum are absorbed by various pigment molecules (like chlorophyll a), causing qualitative and quantitative changes to the scattered, reflected, reemitted, reabsorbed, or otherwise modified spectrum of light coming from the surface of the leaf, measured as reflectance. By using spectral feature variables, monitoring and diagnosing plant responses to the environment can provide important information for production management decisions [24]. Wittenberghe et al. reviewed the application of hyperspectral technology in vegetation photosynthesis at the leaf, tower, airborne, and satellite scales [25]. Adrián Moncholi-Estornell et al., after considering factors such as instrumentation and experimental setup, quantified the variability of plant reflectance in the PRI spectral region (i.e., 500–600 nm) using different laboratory protocols, which is beneficial for characterizing photochemical properties [26]. They also studied the photosynthetic physiological characteristics of leaves and canopies of Salvia farinacea and Datura stramonium plants and proposed a linear spectral unmixing method based on solar-induced chlorophyll fluorescence spectra [27].

Spectral data are abundant, and spectral indices (SI) and other spectral feature variables can be effectively used to enhance certain features or details of vegetation, making statistical analysis results more accurate and convincing [28]. Various spectral preprocessing and modeling methods can also improve the processing speed and accuracy of models. Su Kai and others used spectral remote sensing technology to reflect the leaf spectral characteristics of Euonymus japonicus under the influence of coarse particulate matter. They assessed the impact of dust retention on the prediction accuracy of seven spectral feature variables, including spectral three-edge position, green peak, purple valley, normalized difference vegetation index (NDVI), and enhanced vegetation index (EVI), and concluded that the more particulate matter retained, the poorer the model’s predictive ability [29]. Peiqi Yang and others created a fluorescence-corrected vegetation index (FCVI) to monitor the absorption of photosynthetically active radiation by plants and the processes of chlorophyll fluorescence scattering and reabsorption, enabling accurate and efficient management of vegetation [30]. Kong Lijuan and others constructed an optimal spectral vegetation index by finding the optimal wavelength position, which improved the accuracy of the plant physiological information inversion model under the influence of particulate matter [31]. Xiang Youzhen and others found that the optimal spectral index under 1.5-order differentiation had a higher correlation with soybean LAI, and the constructed prediction model’s accuracy was higher than that of 1st- and 2nd-order integer differentials. They concluded that fractional-order differential transformations can extract gradient information that integer-order differentials cannot represent, highlighting subtle spectral variations and enhancing some weaker spectral features [32]. HONG Y pointed out that fractional-order differentiation, as a mathematical extension of integer-order differentiation, gradually weakens background noise with increasing fractional-order differentiation, while high-frequency noise is amplified, which may reduce the signal-to-noise ratio of spectral information [33]. Walid CHOUARI and others used the NDVI to assess vegetation changes and causes, which has important applications for analyzing degraded vegetation areas and identifying problematic regions [34]. Alexey Stepanov and others used the seasonal time series of NDVI to predict soybean yields early. Four different functions were used to approximate seasonal NDVI curves: Gaussian and DL functions, as well as quadratic and cubic polynomials [35]. An improved modified vegetation presence frequency index (VPF) method was proposed by PU Jing et al. The modified VPF method was based on Moderate-resolution Imaging Spectroradiometer (MODIS) imagery, and it could be established using only a small amount of measured data, which is useful for water quality monitoring on large spatial scales [36]. Somayeh Talebiesfandarani and others introduced the microwave vegetation index (MVI), based on polarization-independent (MVIPB) and time-invariant (MVITB) methods, to derive vegetation optical depth (VOD) and vegetation water content (VWC) based on model simulations at the L-band [37]. Neda Abbasi and others used the VI methods to estimate actual evapotranspiration (ETa) and emphasized the importance of continuity correction based on MODIS for improved performance. Compared with ETc values, the ETa estimates based on MODIS-continuity-corrected Landsat-EVI (EVI2) (EVIMccL and EVI2MccL) performed slightly better across croplands than those of uncorrected Landsat-EVI (EVI2) [38]. Jae-Hyun Ryu and others achieved continuous monitoring of NDVI and photochemical reflectance index (PRI), resulting in improved R^2^ values between extracted vegetation indices and measured indices for paddy rice, barley, and garlic crops. The R^2^ values increased by approximately 0.002~0.004 (NDVI) and 0.065~0.298 (PRI), significantly improving the seasonal signals of vegetation indices [39].

Suspended particulate matter causes a reduction in light intensity both inside and outside greenhouses and enhances light scattering radiation, leading to a decrease in the light energy utilization efficiency and net photosynthetic rate of greenhouse vegetables. Based on the characteristics of Young Chinese cabbage leaves, such as their wavy folds and the presence of trichomes on the leaf surface, this paper analyzes the mechanism by which particulate matter causes changes in the microscopic structure of Young Chinese cabbage leaves. It establishes the relationship between spectral feature variables and the net photosynthetic rate of Young Chinese cabbage. The study explores the optimal method for quantitatively inverting the physiological information of greenhouse vegetables in the presence of particulate matter. This research is of great significance for improving the accuracy of spectral recognition, detection, and information inversion as a means to estimate plant characteristics (like fluorescence) in conditions of dust and particulate matter interference (or contamination).

## 2. Materials and Methods

The tested variety is Young Chinese cabbage (scientific name: *Brassica rapa* L.), belonging to the Brassicaceae family and the Brassica genus. It is a non-heading variety of cabbage, an annual or biennial herbaceous plant that can be cultivated and harvested year-round. It is one of the most important leafy vegetables in China. The leaves of Young Chinese cabbage are alternate, with trichomes on both the upper and lower surfaces. The leaf margin is either entire or serrated, and the leaf surface has wavy folds.

The seeds were sown in a sunlight greenhouse at the School of Biological and Agricultural Engineering, Jilin University, China, and the seedlings were transplanted and allowed to acclimatize for 3 days before being subjected to an artificial particulate matter pollution pot experiment at an outdoor experimental base. The control group (masked as CK) consisted of healthy Young Chinese cabbage (15 plants in total) grown in a pollution-free greenhouse. The experimental group (masked as PM) consisted of Young Chinese cabbage (15 plants in total) that was exposed to continuous particulate matter pollution for 3 h daily inside the experimental greenhouse. Apart from the difference in particulate matter exposure, the water and fertilizer management conditions were the same for both groups. The particulate matter pollution was applied directly by extracting 2000 mL from the artificial particulate generating box and introducing it into the experimental greenhouse until the concentration of PM2.5 reached 500 μg/m^3^. At this point, the air pollution index level was classified as Level 6 [40]. When the air pollution index reaches level 6 (with air quality defined as “severe pollution”), high concentrations of pollutants rapidly increase the risk of respiratory, cardiovascular, and other system diseases, threatening human health. Sensitive groups, such as children, the elderly, and those with chronic illnesses, are more severely affected and are at a higher risk of developing acute conditions or experiencing exacerbation of chronic diseases.

## 3. Data Collection and Processing

Data were collected during the harvesting period of Young Chinese cabbage (on clear days between 9:00 AM and 3:00 PM). For each plant, 7–8 functional leaves were measured, labeled as shown in Figure 1. During data collection, the leaf veins were avoided. A total of 210–240 Young Chinese cabbage leaves were sampled for hyperspectral and net photosynthetic rate data, with the data collection positions being the same for both types. The net photosynthetic rate (Pn, μmol·m^−2^·s^−1^) was measured using a photosynthesis meter (LI-6400XT, LI-COR, Lincoln, NE, USA), while hyperspectral data were obtained using a field spectrometer (Field Spec Hand Held 2, ASD Inc., Boulder, CO, USA), with a measurement range of 325–1075 nm, a sampling interval of 1.4 nm, and a resolution of 3 nm @700 nm. All data in this study represented the average of three repeated measurements.

The microscopic experiments on the leaves were conducted at the National Electrochemical and Spectroscopic Research and Analysis Center in China. The experimental procedure was as follows: The leaves were collected and cut into 3 mm × 3 mm samples, then attached to the observation stage using conductive adhesive. After gold spraying (metal spraying process) treatment, they were prepared as scanning electron microscope samples for microscopic observation [41]. The microscopic images were obtained using a Field Emission Scanning Electron Microscope (ESEM, JEOL JSM-6700F, FEI Company, Hillsboro, OR, USA). The data processing and analysis software used were ViewSpec Pro 8.0, Matlab R2017b, Adobe Illustrator 29.0, and Origin 9.8.

## 4. Results and Discussion

### 4.1. Spectral Characteristics

In order to effectively reduce the complexity of the spectral data and shorten the computation time, spectral curves with significant noise at both ends of the full spectral range (325–1075 nm) of Young Chinese cabbage were excluded. Thus, the spectral range analyzed was limited to 400–900 nm. Figure 2b depicts the spectral data smoothed using the Savitzky-Golay (SG) filter with a window size of 10. From Figure 2a,b, it can be seen that the spectral curves of the control group and the experimental group show consistent overall trends, both displaying the spectral features of “one peak, one valley, one shoulder” (green peak, red valley, and stable reflectance plateau). After being exposed to particulate matter, the physiological information inside Young Chinese cabbage leaves changes, which can be reflected in the hyperspectral curve. Generally speaking, the lower reflectance of stressed plants is mainly due to chlorophyll degradation, which reduces photon absorption. At the same time, damage to cellular structures weakens photon scattering ability, and the decrease in photon transition efficiency further exacerbates the imbalance in light energy utilization. Ultimately, the combined effect of reduced pigments and structural damage leads to a significant decrease in reflectance. Therefore, in the wavelength range before 700 nm, the spectral reflectance of the test group of Young Chinese cabbage leaves is overall lower than that of the control group.

In the range of 490–580 nm, the chlorophyll in Young Chinese cabbage leaves absorbs green light (510–560 nm) weakly, causing the green light to be reflected by the leaves, resulting in a peak on the spectral curve, known as the “green peak” at 550 nm. In the range of 620–760 nm, which corresponds to the visible red light region, Young Chinese cabbage’s photosynthetic activity is strongest and the absorption of light is greatest, forming the reflection valley at 680 nm, i.e., the “red valley”. This is consistent with previous studies on the spectral characteristics of urban vegetation canopies affected by dust deposition [42]. In the 680–760 nm wavelength range, a typical “steep slope” feature of vegetation is observed, where the slope of the reflectance spectrum increases sharply, forming the “red edge effect”. In the range of 760–900 nm, the scattering and reflection of light increase due to the reflection from mesophyll cells and intercellular spaces, as well as the leaf’s water content. The reflectance spectra of both groups of Young Chinese cabbage show a steady peak (i.e., the stable reflectance plateau after 760 nm, also called the red shoulder or NIR shoulder). The effect of particulate matter causes damage to the internal structure of Young Chinese cabbage, weakening photosynthesis, reducing light absorption, and increasing reflection. Therefore, the average height of the red shoulder in the experimental group is higher than that in the control group [43].

### 4.2. Mechanism Analysis of Microscopic Results

In this project, scanning electron microscopy (SEM) was performed on healthy Young Chinese cabbage leaves that were not exposed to particulate matter pollution, as well as on leaves that had been polluted by particulate matter. Figure 3 depicts the microscopic results of the leaves. The microscopic observations of Young Chinese cabbage leaves near the main vein and the deposited particulate matter are shown in Figure 4a–e.

Based on the analysis of the microscopic structure of stomata in the leaves of young Chinese cabbage in Figure 3 and Figure 4, it can be observed that after exposure to particulate matter (PM) pollution, the particles induce oxidative stress responses in plant cells. This is reflected in the absorption of particles into the stomatal pores. The oxidative stress responses generated by the plant produce excessive reactive oxygen species (ROS) that damage the cell’s antioxidant system, impairing cellular structure and function. As a result, the microscopic images of the stomata become blurred, the stomatal aperture shrinks, and the particles block the stomata, affecting the plant’s normal physiological activities [29].

From the analysis of the distribution of deposited particles on the leaf surface in Figure 4b–e, it can be seen that the particle size deposited near the leaf veins of the Young Chinese cabbage is larger compared to the smooth areas of the mesophyll. This suggests that the protrusions and grooves near the leaf veins are more likely to adsorb particles, which are primarily coarse particles (PM2.5–PM10, Dp = 2.5–10 μm) and total suspended particles (TSP, Dp ≤ 100 μm). These particles are intercepted and adsorbed on the surface of the leaf by the leaf epidermal trichomes and the leaf vein protrusions. The increased roughness of the leaf surface due to the trichomes and veins makes it easier for the particles to adhere (Figure 4b–e). On the mesophyll areas of the leaf (the smooth area between the leaf margin and the leaf veins), more fine particles (PM2.5, Dp ≤ 2.5 μm) and ultrafine particles (PM1, Dp ≤ 1 μm) are adsorbed (Figure 4b–e). These two types of particles can deposit on the leaf surface, blocking the stomata and hindering the entry of carbon dioxide and the release of oxygen, thereby affecting the plant’s photosynthesis. This leads to a decrease in the plant’s photosynthetic rate, a reduction in the synthesis of organic matter, and subsequently impedes plant growth and development [44]. Furthermore, plants absorb oxygen and release carbon dioxide through respiration. Fine and ultrafine particles are absorbed into the plant’s stomata and intercellular spaces, where they participate in physiological activities and interfere with gas exchange, thus disrupting normal respiration. This blockage impedes the plant’s energy metabolism, which negatively affects plant growth and its ability to resist stress. Additionally, fine and ultrafine particles contain harmful substances such as heavy metals. For example, metal oxide nanoparticles, as shown in Figure 4e, may bind with proteins and enzymes inside the cells, causing them to lose their activity, leading to potential mechanistic harm [44]. The presence of polar functional groups, such as carboxyl, hydroxyl, and amine groups, can aid in the adsorption and sequestration of particulate matter through chemical bonds with the particles [45]. A schematic diagram of the retention, attachment, and absorption of particulate matter by the leaves of Chinese cabbage is shown in Figure 5. Harmful substances, including particulate matter, enter plant cells and, through extra-plastid and symplastic transport pathways, can damage the plant cell membrane structure, organelles, and genetic material, adversely affecting the plant’s physiological functions.

The first derivative spectral characteristics are closely related to plant physiological ecology and can describe the plant’s pigment status, health condition, and growth vitality. After performing first derivative processing on the original spectrum of young Chinese cabbage, as shown in Figure 6a, the first derivative curve of Chinese cabbage reflects the slope change of the original hyperspectral curve. The variation characteristics of the first derivative spectral curves of the two groups of Chinese cabbage are essentially the same. After averaging and plotting, as shown in Figure 6b, the first derivative values of the experimental group are generally higher than those of the control group, which is consistent with the curvature change pattern of the original spectrum. By extracting spectral characteristic parameters within the blue, yellow, and red light ranges in Figure 6a, these parameters can be used as diagnostic features for plant-related parameter identification. This has broad application significance for predicting plant physiological information indicators. The Physiological Reflectance Index (PRI) [(R550 − R530)/(R550 + R530)] and the Modified Chlorophyll Absorption Ratio Index (MCARI) [(R700 − R670) − 0.2 (R700 − R550) ]/(R700/R670) are indicators that reflect the physiological status of vegetation [43]. Therefore, in this study, PRI and MCARI are chosen as one of the spectral characteristic variables to predict the physiological information of Chinese cabbage affected by particulate matter.

As shown in Figure 6b, by extracting the maximum first derivative values within the spectral ranges of 490–530 nm, 560–640 nm, and 680–760 nm from the first derivative spectral curve in Figure 6a, the blue edge amplitude, yellow edge amplitude, and red edge amplitude can be obtained. The corresponding wavelength positions represent the blue edge position, yellow edge position, and red edge position, respectively. In the experimental group, the red edge position exhibits a typical characteristic of stressed vegetation, namely, a shift toward shorter wavelengths, with an average shift of 3 nm. The red edge position is significantly correlated with the net photosynthetic rate of the leaves. By conducting a correlation analysis between the net photosynthetic rate and the spectrum, sensitive spectral bands can be identified. Therefore, the red edge position and sensitive spectral bands are also chosen as one of the spectral characteristic variables [46]. The spectral characteristic variables used in this study for physiological information inversion are shown in Table 1.

### 4.3. Spectral Preprocessing

In order to reduce the impact of factors such as measurement instrument errors, spectral noise, stray light, and baseline drift on the experimental results, the original spectra of the experimental group, Young Chinese cabbage, were processed using nine spectral preprocessing methods. Table 2 revealed the nine spectral preprocessing methods. These methods include convolution smoothing (Savitzky-Golay smooth, SG), first derivative (FD), second derivative (SD), multiplicative scatter correction (MSC), standard normal variate (SNV), as well as their combinations.

The nine spectral preprocessing methods were applied using the Matlab software, and the reflectance spectral curves of Young Chinese cabbage were obtained, as shown in Figure 7a–i.

In studying the correlation between spectral characteristic parameters and the net photosynthetic rate of young Chinese cabbage leaves under particulate matter pollution, a random selection of 3/4 of the samples was used as the calibration set, while 1/4 was used as the validation set. The net photosynthetic rate of 216 leaves of young Chinese cabbage was measured, and the bar chart of the net photosynthetic rate, obtained after calculating the average, is shown in Figure 8.

Within the 400~900 nm wavelength range, the spectral characteristic parameters of red edge position, PRI, and MCARI were Z-score normalized, and a cubic polynomial model ($Y=\beta_{3}X^{3}+\beta_{2}X^{2}+\beta_{1}X+\beta_{0}$) was used for fitting. From the fitting results of the polynomial inversion model (Table 3), it can be seen that the physiological information model for Young Chinese cabbage at the harvest stage provides good inversion results. Among these, the regression coefficient of determination (R^2^) for the cubic equation is highest for PRI (R^2^ = 0.84542), indicating the best fitting accuracy, followed by MCARI and λr, with R^2^ values of 0.83083 and 0.82061, respectively. The R^2^ values for the three spectral characteristic variables are all greater than 0.8, indicating a high fitting level. The model can serve as a reference for inverting the net photosynthetic rate of young Chinese cabbage under particulate matter pollution.

In the sensitive wavelength range of 570–675 nm, 24 combinations of spectral preprocessing and modeling methods were used to establish an inversion model for the net photosynthetic rate of Young Chinese cabbage under particulate matter pollution. The optimal combination of spectral preprocessing and modeling methods was selected to more accurately choose the most suitable approach for inverting the physiological information of Young Chinese cabbage under particulate matter pollution. The prediction results of all inversion models for the net photosynthetic rate of Young Chinese cabbage in the validation set are shown in Figure 9.

Comparing the model results of 24 inversion methods for net photosynthetic rate, as shown in Figure 10, it can be seen that under particulate matter conditions, the four combination methods—SG + PLS + MSC, SG + PCR + SNV, SG + SD + PLS + MSC, and SG + SD + PLS + SNV—yield better inversion results, with the model determination coefficients (R^2^) all exceeding 0.9. Among these, the optimal combination of spectral preprocessing and modeling methods is SG + SD + PLS + MSC, with a model determination coefficient of R^2^ = 0.9513. This optimal combination will provide valuable reference for research on the impact of particulate matter pollution on leafy vegetables.

## 5. Conclusions

This study conducted a correlation analysis of the physiological and spectral information of young Chinese cabbage under particulate matter (PM) pollution using photosynthetic physiological information detection, spectral technology application, microscopic imaging, and mechanism analysis methods. By applying cubic polynomial and 24 spectral preprocessing and modeling combinations, a spectral characteristic variable model for net photosynthetic rate inversion was established. The results showed that SG + SD + PLS + MSC was the optimal spectral preprocessing and modeling method combination, which can be used to invert the physiological information of young Chinese cabbage, with a model determination coefficient of R^2^ = 0.9513. The findings suggest that harmful substances like particulate matter can enter plant cells through stomata, affecting photosynthesis and physiological functions, thus negatively impacting plant photosynthetic physiology.

The environment and the plants themselves are the main factors influencing plant spectra. When studying the growth environment of greenhouse crops, people often focus on conditions such as water, nutrients, temperature and humidity, pests, and diseases, while overlooking particulate matter. However, particulate matter can affect the light and air environment in the greenhouse, absorb into the plant, and hinder normal physiological activities, making efficient photosynthesis difficult. Therefore, further research on the physiological response mechanisms of plants under particulate matter pollution and spectral inversion will be of significant importance for precise greenhouse environmental control and promoting high-quality plant growth and development. Particulate matter pollution has posed a risk to greenhouse vegetables. Better environmental regulations and legal protections are key to ensuring the sustainable development of agriculture. By legislating to restrict pollution, strengthening technical support, and improving accountability mechanisms, we can effectively resist the threat of particulate matter pollution to vegetable production and protect food safety for humans.

## Figures and Tables

**Figure 1 plants-14-01479-f001:**
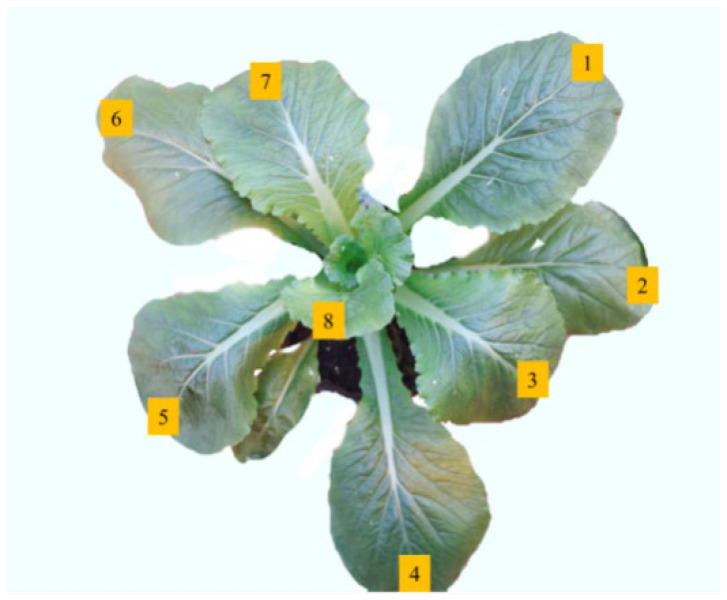
Leaves of Young Chinese cabbage. Note: 1–8 represent the labels for the 8 leaves used for measurement.

**Figure 2 plants-14-01479-f002:**
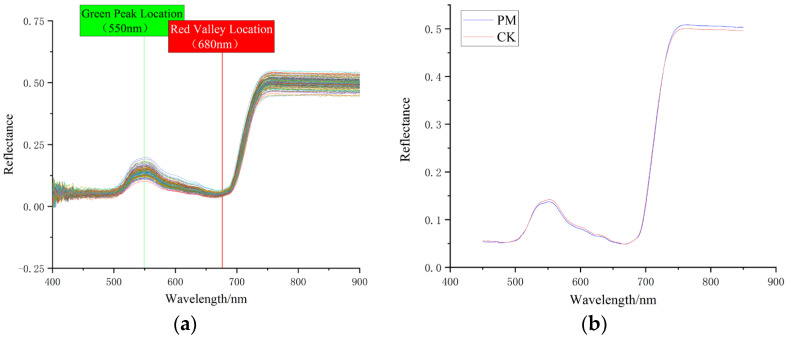
Spectral characteristics of Young Chinese cabbage leaves under PM (PM) and non-PM (CK). (**a**) Original spectrum; (**b**) Spectrum after averaging and SG smoothing.

**Figure 3 plants-14-01479-f003:**
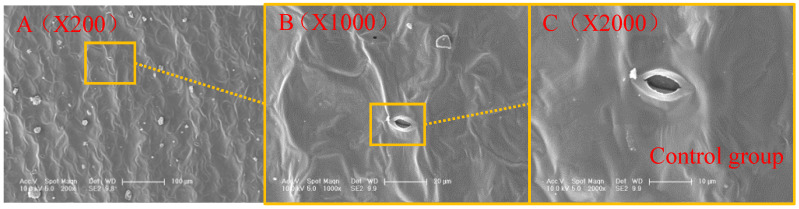
ESEM diagram of healthy young Chinese cabbage leaves. (**A**) shows the middle part of the leaf, specifically the mesophyll area, avoiding the main leaf veins; (**B**) is a local magnified image of the area outlined in (**A**); (**C**) is a further local magnification of the area outlined in (**B**).

**Figure 4 plants-14-01479-f004:**
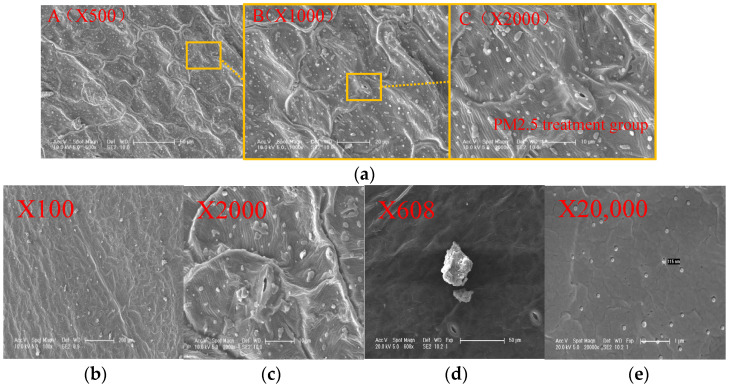
ESEM diagram of young Chinese cabbage leaves under PM. (**A**) shows the middle part of the leaf, specifically the mesophyll area, avoiding the main leaf veins; (**B**) is a local magnified image of the area outlined in (**A**); (**C**) is a further local magnification of the area outlined in (**B**). (**a**) ESEM diagram of young Chinese cabbage leaves under PM; (**b**) the particles on the leaves; (**c**) the particles near the leaf veins; (**d**) the particles between the leaf margin and the leaf veins; (**e**) particles such as metal oxide nanoparticles.

**Figure 5 plants-14-01479-f005:**
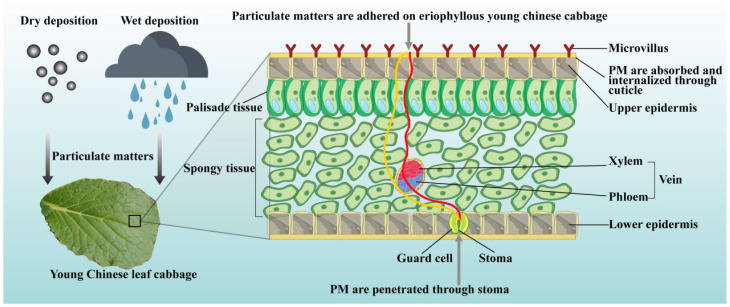
The schematic diagram of retention, adhesion, and absorption of PM by the leaves of young Chinese cabbage.

**Figure 6 plants-14-01479-f006:**
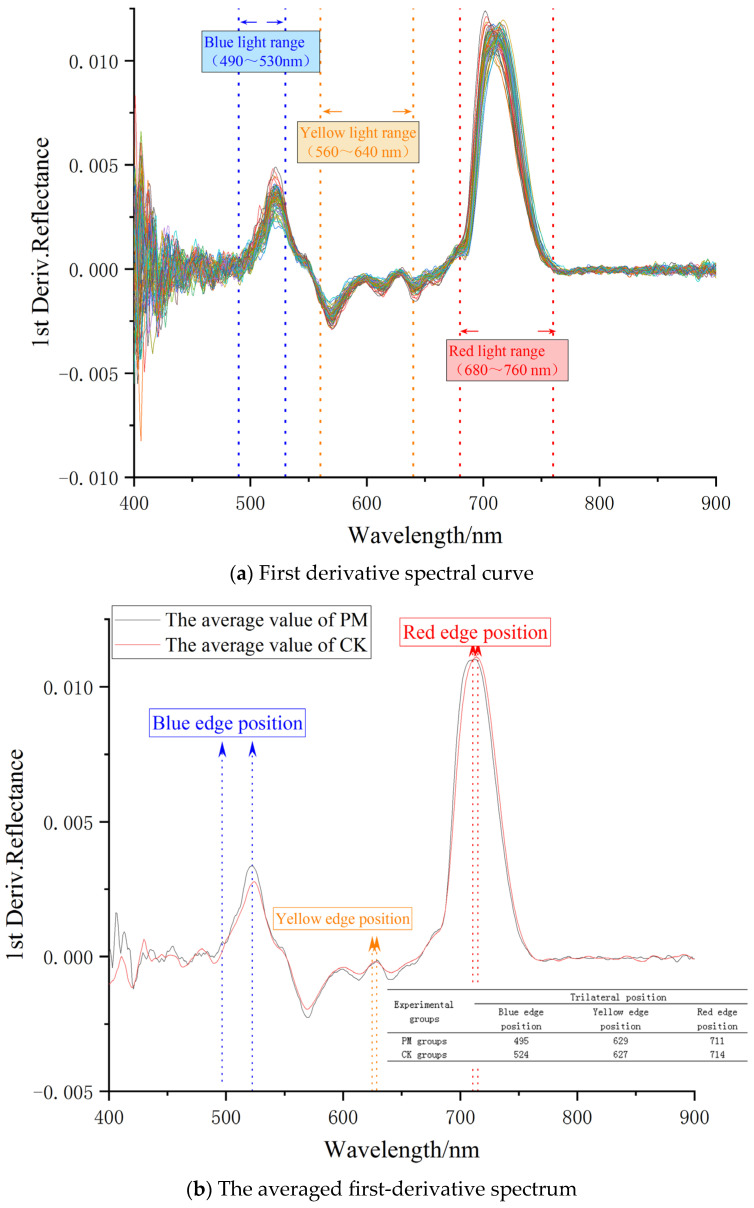
First derivative spectrum.

**Figure 7 plants-14-01479-f007:**
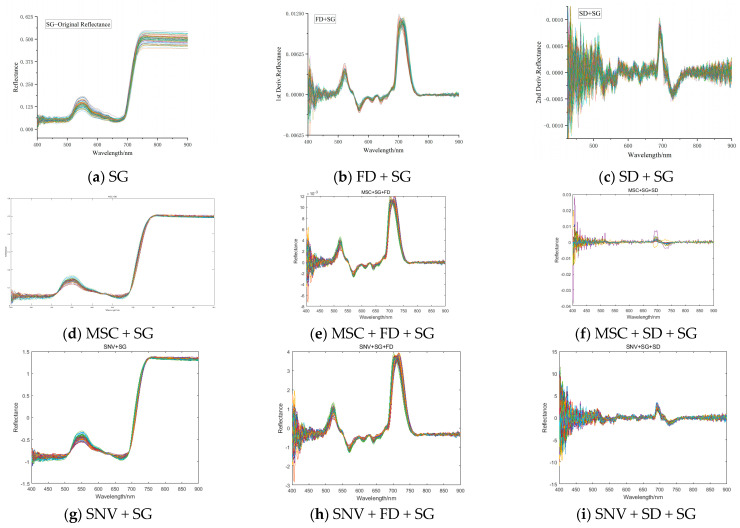
Spectral reflectance curves after nine preprocessing methods.

**Figure 8 plants-14-01479-f008:**
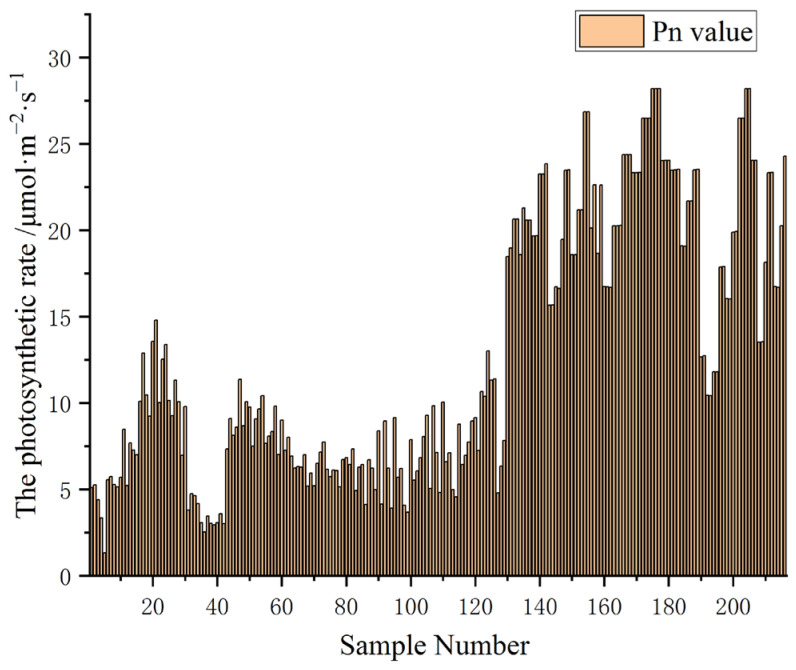
The average net photosynthetic rate of the measured leaves.

**Figure 9 plants-14-01479-f009:**
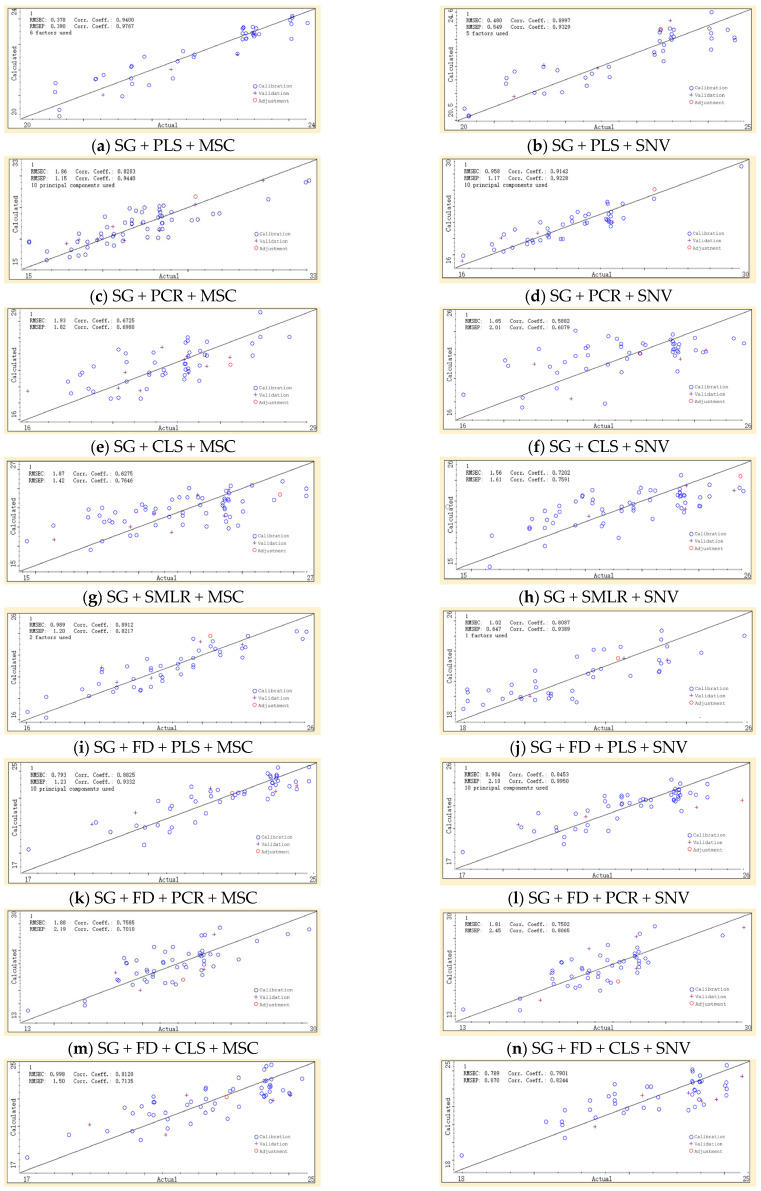
Verification set prediction results of Pn inversion model of young Chinese cabbage.

**Figure 10 plants-14-01479-f010:**
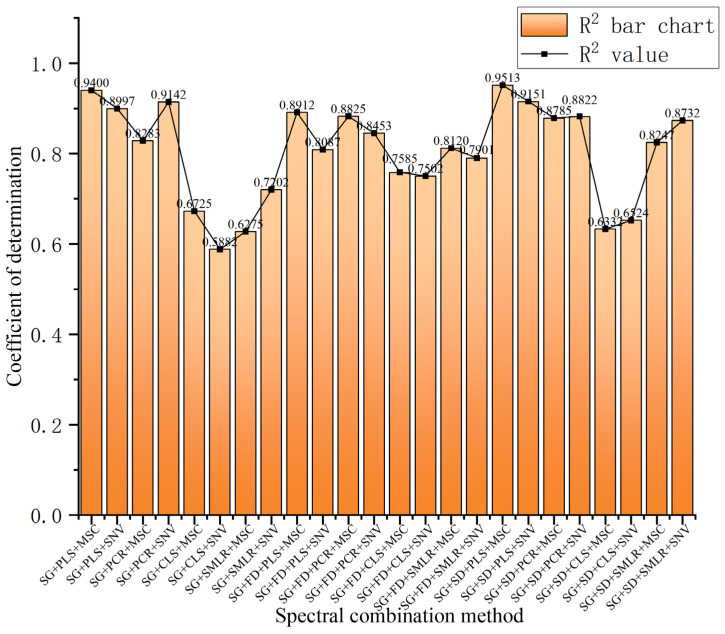
Comparison of modeling results of 24 spectral combination methods.

**Table 1 plants-14-01479-t001:** The spectral characteristic variables.

Spectral Characteristic Variables	Parameter Description
Red edge position (λr)	The wavelength position corresponding to the maximum of the red wavelength range (i.e., the red edge amplitude)
PRI	(R550 − R530)/(R550 + R530)
MCARI	[(R700 − R670) − 0.2 (R700 − R550)]/(R700/R670)
Sensitive spectral bands	The spectral characteristic band (570~675 nm) identified through correlation analysis as sensitive spectral band

**Table 2 plants-14-01479-t002:** Spectral pretreatment method.

Spectral Pretreatment Method	SG	
SG	SG	
FD	FD + SG	
SD	SD + SG	
MSC	MSC + SG	FD + MSC + SG
		SD + MSC + SG
SNV	SNV + SG	FD + SNV + SG
		SD + SNV + SG

**Table 3 plants-14-01479-t003:** Inversion models of Pn and spectral characteristic variables.

Spectral Characteristic Variables	Fitting Model	Determination Coefficient (R^2^)
λr	y = 79.95731 + 87.01537X − 3.965X^2^ + 0.05966X^3^	0.82061
PRI	y = −0.77578 + 0.11797X − 0.00566X^2^ + 9.41219X^3^	0.84542
MCARI	y = −0.16599 − 0.01887X + 8.05999X^2^ − 1.27072X^3^	0.83083

## Data Availability

The data presented in this study are available on request from the corresponding author. The data are not publicly available due to privacy.

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
