# Peer review of "Study on the Photosynthetic Physiological Responses of Greenhouse Young Chinese Cabbage (Brassica rapa L. Chinensis Group) Affected by Particulate Matter Based on Hyperspectral Analysis"

_plants, 2025, doi:10.3390/plants14101479_

Round 1
Reviewer 1 Report
Comments and Suggestions for Authors
Page 2 .This sentence needs to be re written” The effect of electromagnetic radiation also varies, and chemical bonds will vibrate under different electromagnetic radiation levels, causing differences in the emission and absorption of certain wave lengths of spectra. This results in variations in spectral reflectance.”
Perhaps you mean something like ” Specific wavelengths of the visible light spectrum are absorbed by various pigment molecules (like chlorophyll a), causing qualitative and quantitative changes to scattered, reflected, reemitted, reabsorbed, or otherwise modified spectrum of light coming from the surface of the leaf, measured as reflectance. “
Page 4 “This research is of great significance for improving the accuracy of spectral recognition, detection, and information inversion.”
Perhaps you mean ““This research is of great significance for improving the accuracy of spectral recognition, detection, and information inversion as a means to estimate plant characteristics (like fluorescence) in conditions of dust and particulate matter interference (or contamination)”
(scientific name: Brassica rapa L. Chinensis Group).
We write them in cursive font “Brassica rapa L”
“The particulate matter pollution was applied directly by extracting 2000 mL form the artificial particulate generating box and introducing it into the experimental greenhouse until the concentration of PM2.5 reached 500 μg/m³. At this point, the air pol lution index level was classified as Level 638”
Please write one or two sentences about risk to human heath at this level of exposure to PM2.5
In general, high-energy dust particles excited by photons undergo energy level transitions, emitting beams with strong transitions.
This sentence is not clear at all. Dust particles cannot be “high-energy” if they are stationary on the leaf. Dust particles almost certainly do not emit beams of light. Please rewrite it.
In the range of 490–580 nm, this wavelength range corresponds to the period of max imum chlorophyll content in the leaves of leafy vegetables at harvest.
Do you mean that you measured chlorophyll content in this range over some period of time to determine point of maximal value?
Figure1.2
The part presented as inset contains the key information here, and it should be shown in the main body of the figure.
Figure 1.3 should be marked as “control group” and Figure 1.4 should be marked as “ PM2.5 treatment group”
It seems like the whole point of Figure1.5 was to show EM picture of PM2.5. It should be incorporated to figure 1.4 and size of particles should be better described.
Based on the analysis of the microscopic structure of stomata in the leaves of young Chinese cabbage in Figures 1.3 and 1.4, it can be observed that after exposure to particulate matter (PM) pollution, the particles induce oxidative stress responses in plant cells.
IMPORTANT: How did you determine that the plants are in oxidative stress condition?
it can be seen that the particles near the leaf veins have a larger particle
size compared to those at the leaf edge and in the middle of the leaf between the veins.
Please produce statistics confirming that in fact the particles near the leaf veins have a larger particle size compared to those at the leaf edge. At this point it is just speculation.
In conclusions: Maybe it would be good to mention the necessity for better environmental regulations and legal protections?
Reviewer 2 Report
Comments and Suggestions for Authors
Comments paper “Study on the photosynthetic physiological responses of greenhouse young Chinese Cabbage affected by particulate matter based on hyperspectral analysis.”
Kong et al., 2025, Plants, 3568081.
The paper describes the results of a study concerning the impact of pariculate matter on photosynthetic physiological responses in Chinese Cabbage leaves. Measurements of hyperspectral imaging, net photosynthetic rate and microscopic leaf structure were performed. Several spectral feature variables were analysed through cubic polynomial and 24 combinations of spectral pre-processing and modelling methods. The combination of Savitzky-Golay smooth, second derivative, partial least squares and multiple scatter correction delivers the highest coefficient of determination. The results indicate that particular matter affects plant photosynthesis.
Comments:
The paper is well written, the introduction gives already a detailed overview of the literature but references to studies on, for instance, the impact of air pollution on trees and crops by detailed hyperspectral and sun-induced fluorescence are not mentioned. See therefore the papers of José Moreno's group from the University of Valencia, Spain.
The way the indication to the references throughout the whole manuscript is quite careless.
Materials and Methods:
p.4: from “The experimental group ((Masked as PM)…. to the end of this paragraph, more detailed information is needed:
- which particulate matter is involved?
- water and fertilizer management: describe in detail.
Paragraph 3. Data collection and processing: here the measurements of the photosynthetic rate (Pn) is mentioned but in the section “Results and Discussion”, no tables or figures showing these results are presented, also not a reference to a possible earlier study. See also further comments on p7 and 10 and 12 ‘Conclusions’.
The second part of this paragraph starting from “The experimental procedure was as follows…..”: this should be written in a more proper and fluent way and not with arrows in the text. Details of the different stages of the experimental procedure should be in a supplementary file or at least referring to literature.
Results and Discussion.
p.5, paragraph 1.4: in the first part: what are high-energy dust particles, give examples and at least a reference.
In the second part of this paragraph: where are the chlorophyll data?
p.6, fig 1.2: the inset graph is not readable. Make it bigger. I cannot see the difference between PM and CK.
p.6, figs 1.3, 1.4 and 1.5:
- In all the figures (1.3, 1..4 and 1.5, a, b, c) the magnification should be indicated more clearly, cannot read this, unless I magnify the figures in the pdf-file to at least 200%; or mention the magnifications in the legends.
- The "white spots" in fig 1.3 A and in fig 1.4/1.5, are these of the same material? Please specify.
p.7,
- Did you measure the oxidative stress or did you obtain this from literature? If so, mention at least a reference.
- Where are the data of the measurements of the photosynthetic rate and the reduction of organic matter? See also the comment on paragraph 3 and on page 10.
p.8,
- indices: mention the reference in which these indices has been defined.
- fig 1.7: see remark fig 1.2: the figure 1.7 is difficult to read, can hardly see the difference between the PM and CK first derivative.
p.9, “by conducting….”: for a good understanding, these should be discussed in relation to the literature.
p.10, see remark on p 7, where are the data of the photosynthetic rate?
p.11/12, fig 1.9: Figures are unreadable!
p.12, fig 1.10: hard to read, especially the x-axis.
Conclusions:
Where is the physiological information?
General comment: although the paper contains useful information and results, the main problem is that the physiological data are missing and therefore it is quite impossible to evaluate the correlation analysis and the final conclusions. A major revision is absolutely necessary before the manuscript can be considered for publication.
Round 2
Reviewer 2 Report
Comments and Suggestions for Authors
Dear authors,
Thank you for answering profoundly the comments I made on the manuscript. There are still two small remarks:
- p.3: Wittenberghe S.V. should Wittenberghe et al.
- concerning the Pn data: Why not including the figures you sent in your cover letter in the manuscript or in a supplementary file?
I agree to accept the paper after these minor revisions.
